# Burden of rare deleterious variants in WNT signaling genes among 511 myelomeningocele patients

**Luke Hebert[1], Paul Hillman[1], Craig Baker[1¤], Michael Brown[2], Allison Ashley-Koch[3], James E. Hixson[2], Alanna C. Morrison[2], Hope Northrup[1], Kit Sing Au[1]***

**1** Department of Pediatrics, Division of Medical Genetics, McGovern Medical School at The University of Texas Health Science Center at Houston (UTHealth), Houston, TX, United States of America, **2** Human Genetics Center, Department of Epidemiology, Human Genetics and Environmental Sciences, School of Public Health at The University of Texas Health Science Center at Houston (UTHealth), Houston, TX, United States of America, **3** Department of Medicine and Duke Molecular Physiology Institute, Duke University Medical Center, Durham, NC, United States of America

¤ Current address: Munroe-Meyer Institute, University of Nebraska Medical Center, Omaha, NE, United States of America
* Kit-Sing.Au@uth.tmc.edu

**Data Availability Statement:** The data that support the findings of this study are openly available from the Single Nucleotide Polymorphism database (dbSNP), found at https://www.ncbi.nlm.nih.gov/

## Abstract

Genes in the noncanonical WNT signaling pathway controlling planar cell polarity have been linked to the neural tube defect myelomeningocele. We hypothesized that some genes in the WNT signaling network have a higher mutational burden in myelomeningocele subjects than in reference subjects in gnomAD. Exome sequencing data from 511 myelomeningocele subjects was obtained in-house and data from 29,940 ethnically matched subjects was provided by version 2 of the publicly available Genome Aggregation Database. To compare mutational burden, we collapsed rare deleterious variants across each of 523 human WNT signaling genes in case and reference populations. Ten WNT signaling genes were disrupted with a higher mutational burden among Mexican American myelomeningocele subjects compared to reference subjects (Fishers exact test, $P \leq 0.05$) and seven different genes were disrupted among individuals of European ancestry compared to reference subjects. Gene ontology enrichment analyses indicate that genes disrupted only in the Mexican American population play a role in planar cell polarity whereas genes identified in both populations are important for the regulation of canonical WNT signaling. In summary, evidence for WNT signaling genes that may contribute to myelomeningocele in humans is presented and discussed.

## Introduction

Myelomeningocele is the most common neural tube defect (NTD), with a prevalence of 3.2 per 10,000 births [1]. Affected people are born with both meninges and spinal cord exposed through a cleft in their vertebral column. People with myelomeningocele ordinarily survive

snp/, under accession number/BioProject ID: PRJNA611755.

**Funding:** For exome sequencing, the National Institute of Health's Eunice Kennedy Shriver National Institute of Child Health & Human Development awarded Kit-Sing Au (KA) the grant numbered 5R01HD073434 (https://urldefense. proofpoint.com/v2/url?u=https-3A__project reporter.nih.gov_project-5Finfo-5Fdescription. cfm-3Fprojectnumber-3D5R01HD073434- 2D05&d=DwIGaQ&c=bKRySV-ouEg_AT- w2QWsTdd9X__KYh9Eq2fdmQDVZgw&r= VkWBAnv-bUNSziZvFEP1Il8maq8XQJ0rSgEH- 0uBTc0&m=TR2cQ3DpEHWj9jZOxbu PdU9ftRv5wfV4D8TMg_m6wgw&s=IU6EML_ vdznk-0qm7pgyXx66fW2xBT7YpzLVI_7r7v0&e=). For subject recruitment, the National Institute of Health's Eunice Kennedy Shriver National Institute of Child Health & Human Development awarded Hope Northrup (HN) the grant numbered 5P01HD035946 (https://urldefense.proofpoint. com/v2/url?u=https-3A__report.nih.gov_ categorical-5Fspending-5Fproject-5Flisting.aspx- 3FFY-3D2009-26ARRA-3DN-26DCat-3DSpina- 2BBifida&d=DwIGaQ&c=bKRySV-ouEg_AT- w2QWsTdd9X__KYh9Eq2fdmQDVZgw&r= VkWBAnv-bUNSziZvFEP1Il8maq8XQJ0rSgEH- 0uBTc0&m=TR2cQ3DpEHWj9jZOxbu PdU9ftRv5wfV4D8TMg_m6wgw&s=8305I-7yq Es3TXp5AdF2w9l7FKDEGOG0zkGhdytuDlc&e=). LH received travel support for attending the 2019 International Neural Tube Defects Conference at Boston, MA to allow display and discussion of the data with grant number NIH R13 HD 100191. The funding agencies had no role in study design, data collection and analysis, decision to publish, or preparation of the manuscript.

**Competing interests:** The authors have declared that no competing interests exist.

with the appropriate medical care but frequently live with comorbidities such as Chiari malformation type II, sensory and motor issues below the opening, and more [2].

Myelomeningocele is a multifactorial disease, with evidence suggesting genetic susceptibilities play an important contributing role. Although maternal folate deficiency and gestational diabetes are both risk factors for NTDs, not all cases are explained by the environment of the fetus. One study indicates that only 27.6% of myelomeningocele cases can be attributed to known risk factors [3]. In fact, the heritability estimate of myelomeningocele in humans is 0.6 [4]. There is an increasing number of naturally occurring and lab-generated knockout mice with disruption of at least 372 genes exhibiting NTD phenotypes in mouse models [5], illustrating the role that genetic mutation can have on this family of disorders in vertebrates.

NTDs in humans occur when the neural tube fails to close between weeks three and four of gestation. At this crucial time, closure is initiated by convergent extension of neural plate and continued as the neuroepithelium bends to form a tube. Aside from controlling other morphological events across many species [6], the planar cell polarity (PCP) pathway regulates this convergent extension and bending of neural plate during embryogenesis [7]. In humans with NTDs, variants predicted to impair protein function have been found in PCP pathway genes [8]. PCP is one branch of the larger group of WNT signaling pathways [9–11]. Some genes from the WNT signaling pathways outside PCP are also implicated in the development of NTDs in humans [12, 13].

Given previous evidence that genes involved in WNT signaling contribute to NTDs in humans and model organisms, we aimed to comprehensively evaluate rare, likely deleterious, coding variants within all WNT signaling pathway genes. To do so, we leveraged a gene-based mutational burden analysis, which provides the following advantages: it does not require multiplex family data, it lends potentially more power than single-variant approaches, and it has been successfully applied to the publicly-available datasets we chose as references in another study [14]. We hypothesize that genes within the WNT signaling pathways harbor rare deleterious variants (RDVs) that are overrepresented in myelomeningocele subjects.

## Materials and methods

### Subject population and sample collection

Recruitment of myelomeningocele subjects with written informed consent was in accordance with an institutional review board at the University of Texas Health Science Center at Houston and is described by Au *et al*. 2008. The protocol, HSC-MS-00-001, for subject data collection was approved by the Committees for the Protection of Human Subjects at The University of Texas Health Science Center at Houston. Only patients with a diagnosis of isolated, non-syndromic myelomeningocele at birth were eligible for the study [15].

We evaluated exome sequence data from both myelomeningocele and publicly available reference populations to look for genomic variation. Genomic DNA samples were used for exome capture with TargetSeq reagents (Life Technologies, Inc.) based on high density oligonucleotide hybridization of GENCODE annotated coding exons, NCBI CCDS, exon flanking sequences (including intron splice sites), small non-coding RNAs and a selection of miRNA binding sites. After capture, libraries were constructed with addition of barcodes (AB Library Builder, Life Technologies, Inc.). Multiplexed sequencing used the Ion Proton platform (Life Technologies, Inc.) based on proton assays for polymerase sequencing of individual DNA molecules in wells of modified semiconductor chips.

Reference population variant data was retrieved from version 2 of the Genome Aggregation Database (gnomAD) [16]. Specifically, we used gnomAD's 8,556 "control" Admixed American (AMR) exome data that includes Mexican Americans, Puerto Ricans, Medellin Colombians,

and Peruvians as well as gnomAD's 21,384 "control" Non-Finnish European (NFE) exome data. The word "Hispanic" will be used when referring to both Mexican American cases and the AMR references collectively. Likewise, the phrase "European ancestry" will be used when referring to both the European American cases and NFE references together.

## WNT signaling gene lists

To analyze all genes within the WNT signaling pathways, the AmiGO2 web-based tool provided by the Gene Ontology (GO) Consortium was used to retrieve 523 unique *Homo sapiens* genes under WNT signaling GO accession number GO:0198738 as listed in the S1 Table. To evaluate which components of the overarching WNT signaling pathway were affected, we used other GO-specific gene lists including WNT protein secretion (GO:0061355), WNT-related planar cell polarity (GO:0060071), canonical WNT/β-catenin signaling (GO:0060070), and WNT-related calcium modulation (GO:0007223). These gene lists can also be found in the S1 Table [17, 18].

## Analysis overview

Primary input files for the analysis of the genes include exome sequencing variant data in the form of variant call format (VCF) files for the myelomeningocele subject samples from Genome Analysis Toolkit (GATK) sequencing and VCF files for the reference population from version 2 of the Genome Aggregation Database (gnomAD) [16]. Variant calls were filtered based on quality control metrics and annotated for genomic function before variant allele burden in the myelomeningocele cases was compared to that of a reference population (Fig 1). The steps of the analysis were largely performed using custom scripts written in Python 2.7 and R 3.5.3.

## Annotation with dbNSFP

Annotation of those input VCF variants was performed using functional predictions from the database of non-synonymous single-nucleotide variant functional predictions (dbNSFP) version 4.0b1a [19] and exon start/stop locations were retrieved from the University of California Santa Cruz (UCSC) table browser [20].

## Quality control and selection of damaging rare variants

The quality control filters for case exome data were chosen to closely match the criteria published by gnomAD. They can be categorized as being focused on variant-specific parameters and sample-specific parameters.

The Genome Analysis Toolkit (GATK) provides many variant-specific parameters. Variants retained for analysis include those with a GATK variant quality score recalibration value of "PASS", root mean squared of mapping quality (MQ) $\geq$ 20, and inbreeding coefficient < -0.3. Of the single nucleotide variants (SNVs) filtered from the WES data, we evaluated only those SNVs that lay within any coding transcript or splice sites of the 523 genes of interest. In order to ensure that the allele counts compared at each locus (single nucleotide position) are representative of each population, we only compared SNVs that have $\geq$ 89.5% coverage of DP $\geq$ 10 in both the myelomeningocele cases and the gnomAD population used as references.

The remaining quality control filters target the quality of a variant within an individual sample. We kept samples that met the following at each locus: a genotype with alternate allele depth $\geq$ 25%, a read depth $\geq$ 10, and genotype quality score $\geq$ 20. A variant site is considered "covered" if its position had a DP $\geq$ 10 in 89.5% of samples.

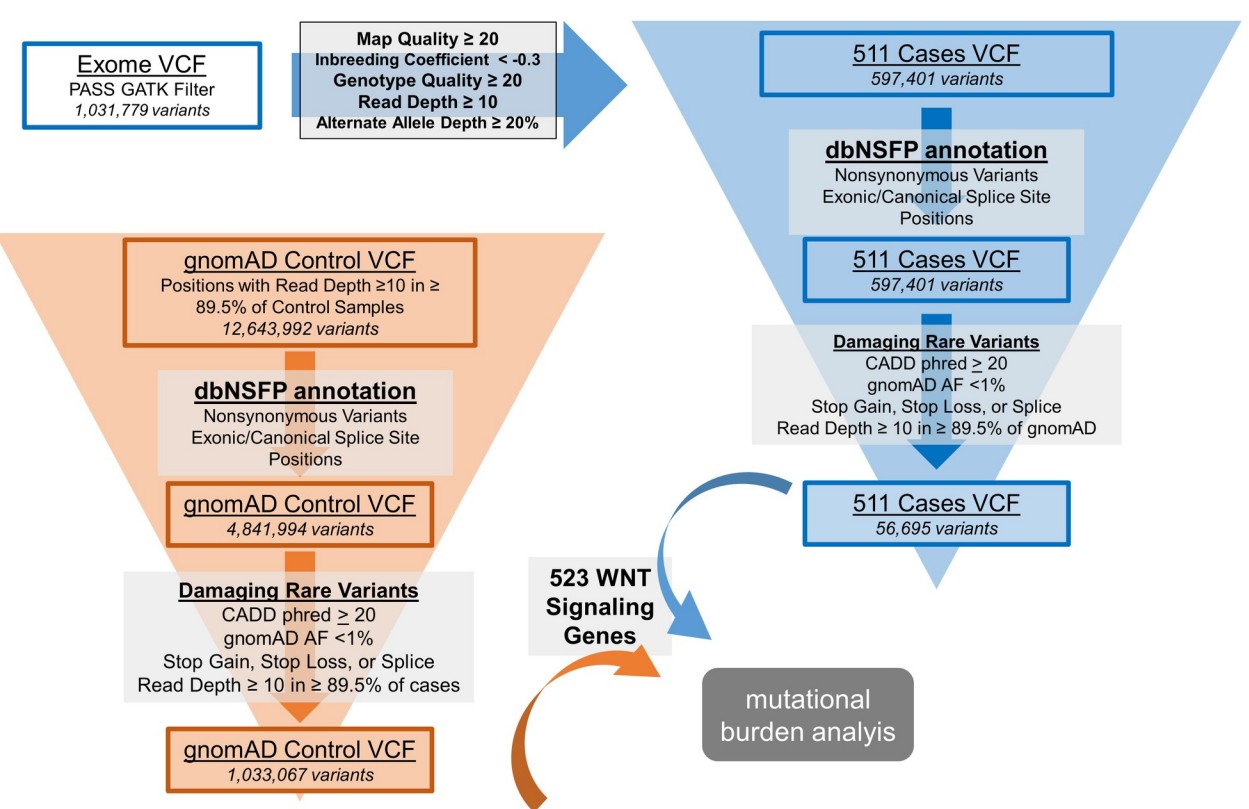

**Fig 1. Variant filtering and annotation workflow.** The analysis pipeline included filtering variants based on sample and variant-level criteria for quality control. The pipeline also included annotating variants with gene names, reference population frequencies, and CADD phred scores so that variant counts could be collapsed over a gene's coding region and so that the analysis could focus on likely deleterious, rare variants. Abbreviations: VCF = variant call format, GATK = Genome Analysis Toolkit, dbNSFP = Database of Non-synonymous Single-nucleotide Variant Functional Predictions version 4.0b1a, gnomAD = Genome Aggregation Database version 2.

In addition to evaluating only high quality variants and samples in the case population, the analysis was further focused to include only variants that were rare (defined as having an alternate allele frequency in the gnomAD reference population less than 1%) and predicted to be deleterious (coding for a stop gain, stop loss, splice site missense, or having a combined annotation dependent depletion phred score greater than 20) [21]. This applies to both myelomeningocele cases and gnomAD references. We refer to rare, deleterious variants that met the above quality control standards as "qualifying variants."

## Mutational burden analysis

To find any genes with higher mutational burden in a case population compared to its gnomAD reference, a two-by-two table for each gene was constructed for Fisher's exact test. The Fisher's exact test compares the number of affected individuals and unaffected individuals from case and the reference populations, where an "affected individual" is someone containing at least one RDV in the gene being evaluated and an "unaffected individual" is someone containing zero RDVs in that gene.

Individual subject data was not readily available from gnomAD. Therefore, affected and unaffected reference population numbers for each gene were estimated using the Hardy-Weinberg equations which utilize the number of qualifying RDVs and the total allele number within a given gene across the gnomAD population. We tested this estimation approach by first

applying it to our case population, generating Q-Q plots that all included a λ value of 2. Any λ value greater than 1 indicates overestimation of affected individuals. For the actual burden analysis, we only applied this estimation to our reference data and not our case data. So, this test indicates that our estimation approach errs on the side of overestimating affected individuals in the reference population. This one-sided overestimation of affected individuals makes our analysis less likely to suggest falsely high mutational burden for genes in the case population.

A Bonferroni correction for multiple comparisons was applied in order to find any genes with significant mutational burden in the case populations. For the Bonferroni correction, the conventional alpha value 0.05 was divided by the number of genes that were compared in a population's mutational burden analysis. The resulting Bonferroni value is a strict *P* value cut-off for statistical "significance". It is important to note that genes were only compared if they harbored qualifying variants, so not all 523 original genes were ultimately compared. Regardless of the Bonferroni correction, the term "nominally significant" refers to genes with Fisher's exact test *P* values ≤ 0.05.

### Gene ontology enrichment analysis

An enrichment analysis of Gene Ontology (GO) terms was conducted using ToppCluster [22] to compare patterns of known biological function between the genes disrupted in Mexican-American (MA) subjects versus genes disrupted in European ancestry (EA) subjects.

## Results

Of the 523 WNT signaling genes, 173 contained qualifying RDVs (GATK PASS, AF < 1%, CADD phred ≥ 20, coverage ≥ 90%) in the MA myelomeningocele population (S2 Table). When comparing RDV mutational burden in the 173 genes between the two Hispanic populations, ten genes associated with a risk for myelomeningocele by yielding Fisher's exact test *P* values below 0.05 and odds ratios above 1 (Table 1). This included *PORCN*, *CDH2*, *PRICKLE2*, *CPE*, *FUZ*, *PTPRU*, *PSMD3*, *TNRC6B*, *PPP2R1A*, and *FERMT2*. Of these, *PORCN*, *CPE*, and *TNRC6B* were detected caudally in the closing neural tube in humans [23]. *PORCN*, *CDH2*, and *FUZ* have been previously associated with NTD phenotypes in mouse models [24–27]. This analysis gave a Bonferroni correction value of 2.9 x 10$^{-4}$, which none of the genes' *P* values fell below.

In the EA myelomeningocele population, 189 genes contained qualifying variants (S3 Table). After comparing those 189 genes between the case and reference populations of European ancestry, seven genes were associated with a risk for myelomeningocele by giving *P* values below 0.05 and an odds ratio above 1 (Table 1). These genes included *DDB1*, *SDC1*, *CSNK1G2*, *SOSTDC1*, *PLCB1*, *DVL2*, and *TLE3*. Of these, *DDB1*, *PLCB1*, and *DVL2* were detected in the human neural tube during closure and *DVL2* is also associated with NTDs in mouse models [23, 28]. This analysis gave a Bonferroni correction value of 2.6 x 10$^{-4}$, which none of the genes' *P* values fell below.

All patient samples that harbored qualifying RDVs in the seventeen associated genes were heterozygous for those variants, except one MA patient who was hemizygous for his variant in *PORCN*. Subjects from both ethnicities tended to have a similar number of genes that contained qualifying variants, but MA subjects tended to have more nominally significant disrupted genes (Fig 2). Nominally significant disrupted genes were identified in 46 (18%) EA and 66 (26%) MA myelomeningocele subjects. In addition, four combinations of nominally significant disrupted genes, each two genes in length, were found in individual samples from the MA myelomeningocele population. These combinations included *CPE* and *PORCN*,

**Table 1. Mutation burden analysis of WNT signaling genes in MA and EA populations.**

| gene | ethnicity | cases with | cases without | case RDVs | refs with | refs without | ref RDVs | P | OR | CI | NT expression |
|---|---|---|---|---|---|---|---|---|---|---|---|
| PORCN† | MA | 3 | 251 | 3 | 11 | 8545 | 25 | 6.81E-03 | 9.28 | 2.2–34.54 | detected |
| CDH2† | MA | 8 | 246 | 8 | 102 | 8454 | 85 | 1.40E-02 | 2.69 | 1.19–5.54 | not detected |
| PRICKLE2 | MA | 6 | 248 | 4 | 65 | 8491 | 70 | 1.62E-02 | 3.16 | 1.32–7.41 | not detected |
| CPE | MA | 8 | 246 | 7 | 106 | 8450 | 40 | 1.70E-02 | 2.59 | 1.15–5.31 | detected |
| FUZ† | MA | 8 | 246 | 4 | 115 | 8441 | 29 | 2.56E-02 | 2.39 | 1.06–4.86 | not detected |
| PTPRU | MA | 11 | 243 | 9 | 187 | 8369 | 159 | 3.09E-02 | 2.03 | 1.02–3.75 | not detected |
| PSMD3 | MA | 6 | 248 | 3 | 78 | 8478 | 54 | 3.40E-02 | 2.63 | 1.1–6.06 | not detected |
| TNRC6B | MA | 12 | 242 | 8 | 218 | 8338 | 149 | 4.29E-02 | 1.9 | 1.04–3.47 | detected |
| PPP2R1A | MA | 3 | 251 | 3 | 25 | 8531 | 33 | 4.56E-02 | 4.08 | 1.04–13.6 | not detected |
| FERMT2 | MA | 6 | 248 | 5 | 85 | 8471 | 67 | 4.73E-02 | 2.41 | 1.01–5.52 | not detected |
| DDB1 | EA | 8 | 249 | 8 | 156 | 21228 | 67 | 7.88E-04 | 4.37 | 1.96–9.07 | detected |
| SDC1 | EA | 5 | 252 | 4 | 76 | 21308 | 14 | 2.79E-03 | 5.56 | 2.12–13.44 | not detected |
| CSNK1G2 | EA | 9 | 248 | 2 | 280 | 21104 | 21 | 7.91E-03 | 2.74 | 1.34–5.43 | not detected |
| SOSTDC1 | EA | 3 | 254 | 2 | 55 | 21329 | 19 | 3.16E-02 | 4.58 | 1.21–14.41 | not detected |
| PLCB1 | EA | 10 | 247 | 6 | 412 | 20972 | 86 | 3.61E-02 | 2.06 | 1.07–3.88 | detected |
| DVL2† | EA | 5 | 252 | 4 | 156 | 21228 | 81 | 4.34E-02 | 2.7 | 1.04–6.55 | detected |
| TLE3 | EA | 6 | 251 | 2 | 211 | 21173 | 46 | 4.55E-02 | 2.4 | 1.02–5.45 | not detected |

Seventeen gene met nominal significance ($P < 0.05$) in the Hispanic (top of table) or European ancestry (bottom of table) burden analysis. $P$ values were calculated using a Fisher's exact test. Human neural tube expression status was annotated with data provided by the authors of Krupp et al. 2012 (42). Abbreviations are as follows: "EA" = European Ancestry, "MA" = Mexican-American, "cases with" = number of myelomeningocele subjects with a qualifying rare variant in this gene, "cases without" = number of myelomeningocele subjects without a qualifying variant in this gene, "case RDVs" = number of qualifying variants within this gene in the myelomeningocele population, "refs with" = estimated number of gnomAD references with a qualifying rare variant in this gene, "refs without" = estimated number of gnomAD references without a qualifying rare variant in this gene, "ref RDVs" = number of qualifying variants within this gene in the gnomAD reference population, $P$ = likelihood that the distribution occurred by chance, OR = odds ratio, CI = confidence interval, NT = neural tube. †Associated with NTD phenotype in mouse model (s).

*PTPRU* and *PSMD3*, *PTPRU* and *FERMT2*, *FERMT2* and *PSMD3*, *PPP2R1A* and *TNRC6B* (S6 Table). No such combinations were found in individual samples from the EA myelomeningocele population.

The disrupted genes that associated with EA myelomeningocele subjects are different from disrupted genes associated with MA myelomeningocele subjects, i.e. the nominally significant genes from each comparison did not overlap.

Gene Ontology analysis regarding biological processes of the disrupted genes in both populations reveals that both groups have disrupted genes enriched for regulation of WNT signaling and negative regulation of canonical WNT signaling. However, disrupted genes in only the EA subjects were enriched for genes associated inositol phosphate metabolism, disassembly of the destruction complex, and recruitment of AXIN to the membrane (Table 2).

## Discussion

Exomes of the myelomeningocele cases compared to gnomAD references revealed nominally significant disruption among genes that function in WNT trafficking, PCP, WNT/β-catenin signaling, and WNT/Ca$^{2+}$ signaling. Our reference value estimation method overestimates the number of RDVs in the gnomAD reference population, so protective variants are likely also overestimated. Therefore, we focus on genes with an odds ratio greater than one, which indicates a risk for myelomeningocele. To remain conservative in our conclusions, we do not evaluate genes whose mutational burden indicates a protective effect, marked by an odds ratio less

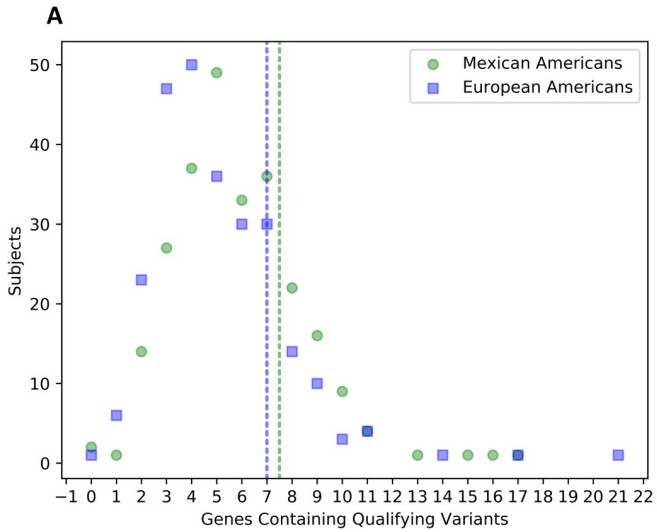
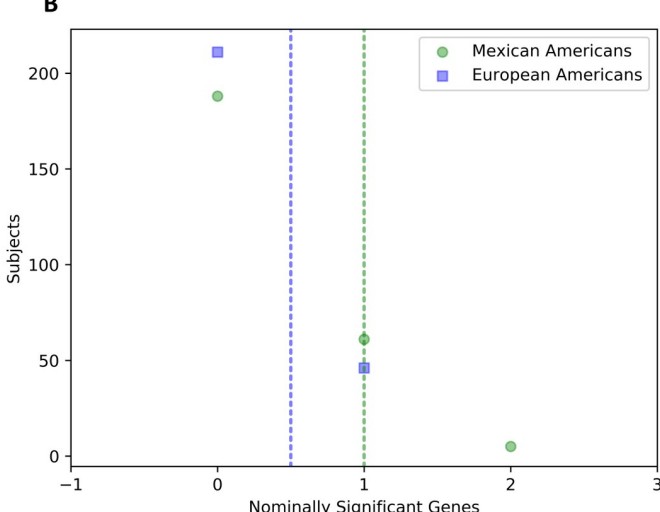

**Fig 2. Distribution of affected genes per subject.** This figure compares Mexican-American (MA) myelomeningocele subjects to European ancestry (EA) myelomeningocele subjects by illustrating how many disrupted genes each subject possessed. Green circles represent MA myelomeningocele data and blue squares represent EA myelomeningocele data. (A) The horizontal axis measures the number of genes that are disrupted by qualifying variants. The vertical axis depicts the number of subjects who possess that number of disrupted genes. (B) The horizontal axis counts only the number of genes that indicated nominally significant risk in the mutational burden analysis. The vertical axis shows the number of subjects with that number of disrupted genes. The vertical dotted lines from either graph represent the median number of disrupted genes possessed by subjects, with green lines representing MA and blue representing EA medians. Both populations followed a similar distribution for how many genes are disrupted per subject, complete with a similar median. However, the Mexican Americans tended to have more nominally significant disrupted genes.

than one. All nominally significant genes with an odds ratio greater than one are included in Table 1, but our discussion further focuses on ten genes that have either been associated with NTD phenotypes, were expressed during closure of the caudal human neural tube, or harbored individual variants that were much more common in the cases than the reference populations. Nominally significant genes meeting one or more of these criteria are: *PORCN*, *CPE*, *FUZ*, *TNRC6B*, *DVL2*, *DDB1*, *PRICKLE2*, *CDH2*, *PPP2R1A*, and *SOSTDC1*. For clarity, we organize each of these ten genes based on their known role in WNT trafficking (Fig 3), the PCP pathway (Fig 4), the canonical β-catenin pathway (Fig 5) or noncanonical Ca2+ WNT signaling pathway (S1 Fig). These figures provide visual references for these pathways with special attention drawn to the translational products of the ten disrupted genes mentioned above [29].

## Myelomeningocele exomes with WNT trafficking genes disrupted

Disrupting the process of WNT synthesis or secretion has the potential to directly and indirectly inhibit one or more of the three downstream WNT signaling pathways. WNT ligands are expressed, processed, and secreted into extracellular space before finding their target cell membrane, where they can bind several cell surface receptors [30] (Fig 3).

In the Hispanic mutational burden comparison, the X chromosome gene *PORCN* associated with risk for myelomeningocele. *PORCN* is necessary for the post-translational modification of the WNT3A ligand. *PORCN* belongs to an evolutionarily conserved gene family termed "Porcupine," whose members code for Wnt processing proteins across species [31, 32]. In humans, the PORCN protein catalyzes O-palmitoylation of WNT3A's Ser209 residue, allowing WNT3A to leave the endoplasmic reticulum [33, 34] with the help of Wntless/WLS [35]. Therefore, the likely deleterious variants found in *PORCN* in our MA myelomeningocele population may prevent WNT3A from leaving the endoplasmic reticulum, ultimately downregulating all WNT signaling pathways in the target cell because less WNT3A would be secreted.

**Table 2. Gene Ontology (GO) enrichment analysis.**

| Category (ID) | Title (or Source) | EA -logP | MA -logP | EA Gene Set | MA Gene Set |
|---|---|---|---|---|---|
| Biological Process (GO:0001736) | establishment of planar polarity | 0 | 4.5 | | FUZ,PRICKLE2,PSMD3 |
| Biological Process (GO:0001738) | morphogenesis of a polarized epithelium | 0 | 4.3 | | FUZ,PRICKLE2,PSMD3 |
| Biological Process (GO:0007164) | establishment of tissue polarity | 0 | 4.49 | | FUZ,PRICKLE2,PSMD3 |
| Biological Process (GO:0016055) | Wnt signaling pathway | 10 | 10 | CSNK1G2,DDB1,DVL2, PLCB1,SDC1,SOSTDC1, TLE3 | CDH2,CPE,FERMT2,FUZ, PORCN,PPP2R1A,PRICKLE2, PSMD3,PTPRU,TNRC6B |
| Biological Process (GO:0022603) | regulation of anatomical structure morphogenesis | 0 | 4.94 | | CDH2,CPE,FERMT2,FUZ, PRICKLE2,PSMD3 |
| Biological Process (GO:0030111) | regulation of Wnt signaling pathway | 5.38 | 10 | CSNK1G2,DVL2, SOSTDC1,TLE3 | CDH2,FUZ,PPP2R1A,PSMD3, PTPRU |
| Biological Process (GO:0030178) | negative regulation of Wnt signaling pathway | 4.36 | 5.56 | DVL2,SOSTDC1,TLE3 | CDH2,FUZ,PSMD3,PTPRU |
| Biological Process (GO:0035567) | non-canonical Wnt signaling pathway | 0 | 4.3 | | PRICKLE2,PSMD3,TNRC6B |
| Biological Process (GO:0045445) | myoblast differentiation | 5.44 | 0 | PLCB1,SDC1,SOSTDC1 | |
| Biological Process (GO:0060070) | canonical Wnt signaling pathway | 10 | 10 | CSNK1G2,DVL2,SDC1, SOSTDC1,TLE3 | CDH2,FUZ,PORCN,PSMD3, PTPRU |
| Biological Process (GO:0060828) | regulation of canonical Wnt signaling pathway | 5.8 | 5.04 | CSNK1G2,DVL2, SOSTDC1,TLE3 | CDH2,FUZ,PSMD3,PTPRU |
| Biological Process (GO:0090090) | negative regulation of canonical Wnt signaling pathway | 4.6 | 5.88 | DVL2,SOSTDC1,TLE3 | CDH2,FUZ,PSMD3,PTPRU |
| Biological Process (GO:0198738) | cell-cell signaling by wnt | 10 | 10 | CSNK1G2,DDB1,DVL2, PLCB1,SDC1,SOSTDC1, TLE3 | CDH2,CPE,FERMT2,FUZ, PORCN,PPP2R1A,PRICKLE2, PSMD3,PTPRU,TNRC6B |
| Coexpression (M7672) | Genes down-regulated in the in vitro follicular dendritic cells from peripheral lymph nodes (96h): Pam2CSK4 versus tretinoin [PubChem = 444795]. | 4.68 | 0 | CSNK1G2,PLCB1,TLE3 | |
| Drug (CID000001233) | ibotenic acid | 0 | 4.76 | | CPE,PORCN,PSMD3 |
| Drug (CID000350833) | purine riboside 5'-monophosphate | 0 | 5.11 | | PORCN,PSMD3 |
| Interaction (int: BBS12) | BBS12 interactions | 0 | 5.24 | | FERMT2,PPP2R1A |
| Interaction (int: DCAF16) | DCAF16 interactions | 4.31 | 0 | CSNK1G2,DDB1 | |
| Pathway (1269487) | Signaling by SCF-KIT | 0 | 4 | | PPP2R1A,PSMD3,PTPRU, TNRC6B |
| Pathway (1269594) | Signaling by Wnt | 4.75 | 4 | CSNK1G2,DVL2, PLCB1,TLE3 | PORCN,PPP2R1A,PSMD3, TNRC6B |
| Pathway (1269599) | TCF dependent signaling in response to WNT | 3.62 | 0 | CSNK1G2,DVL2,TLE3 | |
| Pathway (1269601) | Disassembly of the destruction complex and recruitment of AXIN to the membrane | 3.88 | 0 | CSNK1G2,DVL2 | |
| Pathway (PW:0000154) | inositol phosphate metabolic | 4.21 | 0 | CSNK1G2,PLCB1 | |
| Pubmed (17041588) | CUL4-DDB1 ubiquitin ligase interacts with multiple WD40-repeat proteins and regulates histone methylation. | 4.96 | 0 | DDB1,TLE3 | |
| Pubmed (18351662) | WNT signaling affects gene expression in the ventral diencephalon and pituitary gland growth. | 4.62 | 0 | DVL2,SOSTDC1 | |

*(Continued)*

**Table 2.** (Continued)

| Category (ID) | Title (or Source) | EA -logP | MA -logP | EA Gene Set | MA Gene Set |
|---|---|---|---|---|---|
| Pubmed (23153495) | Sox2 in the dermal papilla niche controls hair growth by fine-tuning BMP signaling in differentiating hair shaft progenitors. | 5.46 | 0 | SDC1,SOSTDC1 | |
| Pubmed (28619992) | Patterning and gastrulation defects caused by the tw18 lethal are due to loss of Ppp2r1a. | 0 | 5.13 | | CDH2,PPP2R1A |
| Pubmed (30389854) | WNT/Î²-catenin signaling plays a crucial role in myoblast fusion through regulation of nephrin expression during development. | 0 | 5.02 | | CDH2,FERMT2 |

These GO terms were enriched in disrupted genes from both Hispanic and European ancestry mutational burden analyses. All results met ToppCluster's Bonferroni correction for the enrichment analysis. "EA" is European Ancestry. "MA" is Mexican American. Gene names are the current symbols recommended by Human Genome Organization (HUGO) Gene Nomenclature Committee. A–logP value of 0 represents a P value above the Bonferroni corrected value.

Previously, *PORCN*'s potential role in the development of NTDs has been suggested in a mouse study where heterozygous constitutive inactivation of its homolog *Porcn* caused open neural tubes *in utero* [24]. Myelomeningocele has been documented in a person with focal dermal hypoplasia (FDH), a rare congenital disorder associated with mutations in *PORCN* [36]. Furthermore, *PORCN* is differentially expressed in the caudal human neural tube during

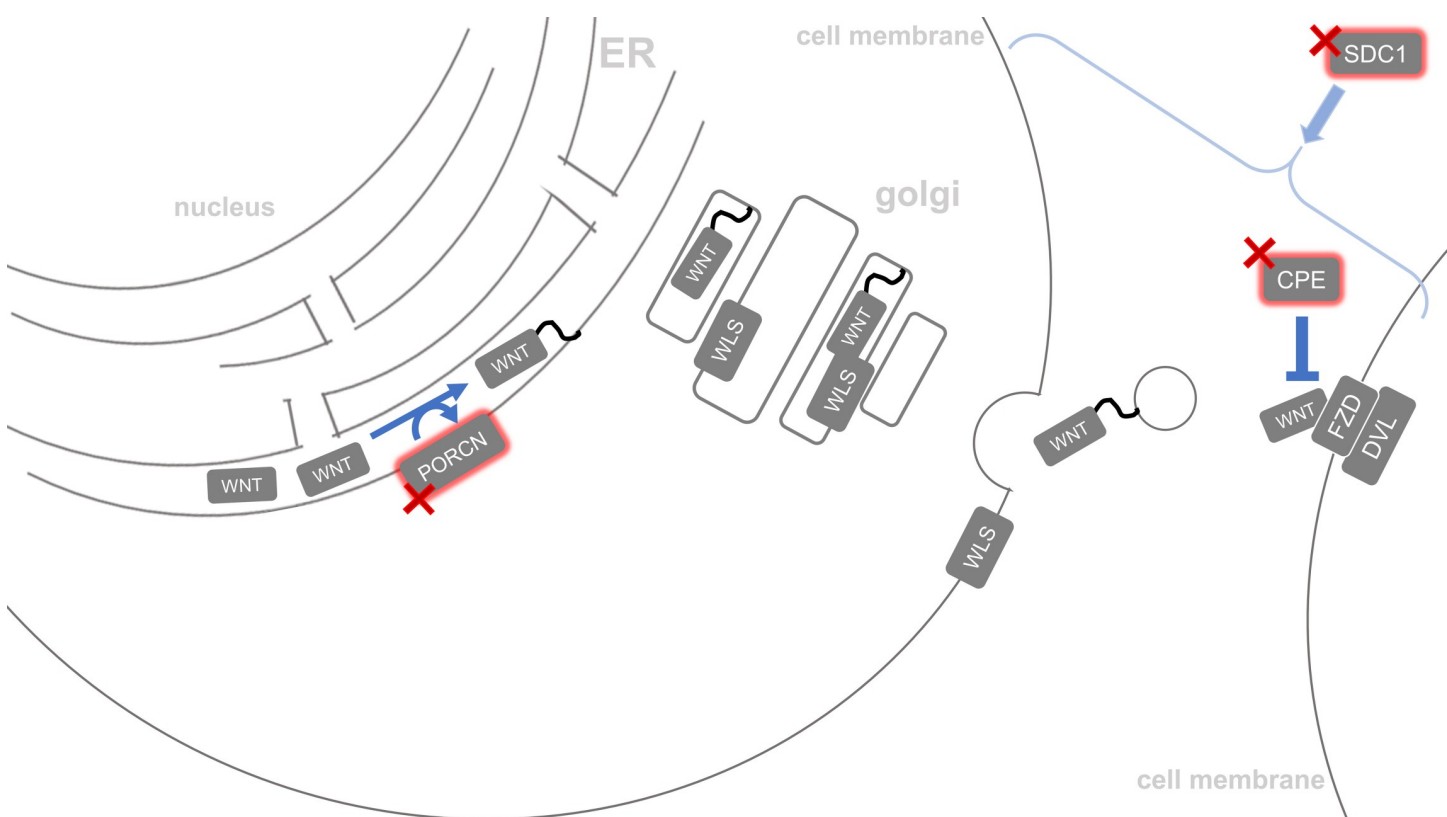

**Fig 3. Altered WNT trafficking.** The proteins PORCN, SDC1, and CPE are all involved in WNT ligand trafficking. Deleterious variants in PORCN may prevent WNT's acetylation which is necessary for WNT to leave the endoplasmic reticulum. Loss of PORCN function would stop WNT from leaving the cell. CPE prevents WNT from binding cell surface receptors on the target cell. Loss of CPE function may indirectly increase WNT's effect on the target cell. Proteoglycans like SDC1 have been implicated as regulators in WNT distribution, though the exact mechanism is not yet known.

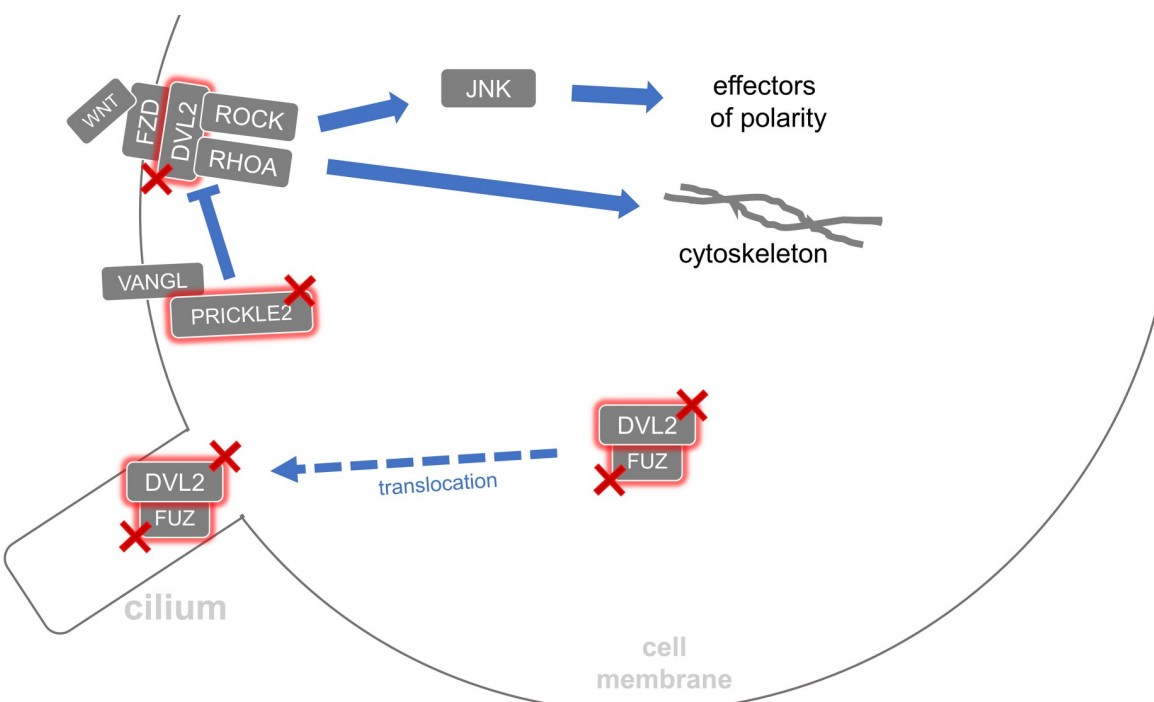

**Fig 4. Altered planar cell polarity.** *DVL2*, *PRICKLE2*, and *FUZ* genes were each disrupted in the myelomeningocele populations. The FZD-DVL complex is necessary to establish planar cell polarity. DVL proteins are also needed to translocate FUZ to cilia. FUZ is essential for the development of a cilium. PRICKLE inhibits the formation of the FZD-DVL complex.

closure [23]. Three myelomeningocele subjects (two females and one male) carried the three RDVs NC_000023.10:g.48371088G>A p.(Gly223Ser), NC_000023.10:g.48371104G>A p.(Arg228His), and NC_000023.10:g.48374165C>G p.(Ala336Gly), all in highly conserved functional regions of *PORCN* [37]. The male who carries NC_000023.10:g.48371088G>A may constitute a homozygote loss of PORCN function. Although a p.(Arg228Cys) has been described as benign in FDH subjects, the NC_000023.10:g.48371104G>A p.(Arg228His) variant we report is different because it is predicted to be damaging by multiple functional analysis algorithms [38]. The NC_000023.10:g.48374165C>G p.(Ala336Gly) variant is located within one of the transmembrane domains of PORCN that forms the substrate transportation pore to provide substrate for acylation of WNT in the ER lumen. The NC_000023.10:g.48374165C>G p.(Ala336Gly) variant is located in between several FDH variants that cause amino acid substitutions such as p.(Leu331Arg), p.(Ser337Arg) and p.(Ala338Pro) [39].

The carboxypeptidase E (*CPE)* gene on chromosome 4, which may affect binding of WNT3A to the target cell's receptor, associated with myelomeningocele in the MA population. The peptide encoded by full-length *CPE* interacts with WNT3A and its receptor Frizzled (FZD) to decrease WNT/β-catenin signaling in a proteasome-dependent manner in human cells [40]. Disruption of this protein's function, therefore, may upregulate the WNT/β-catenin pathway.

## Myelomeningocele exomes with PCP genes disrupted

Closure of the developing neural tube occurs via a process of convergent extension that is orchestrated by the PCP pathway [7]. Any disruption of PCP has the potential to prevent proper neural tube closure. When WNT ligand binds only the cell surface receptor Frizzled

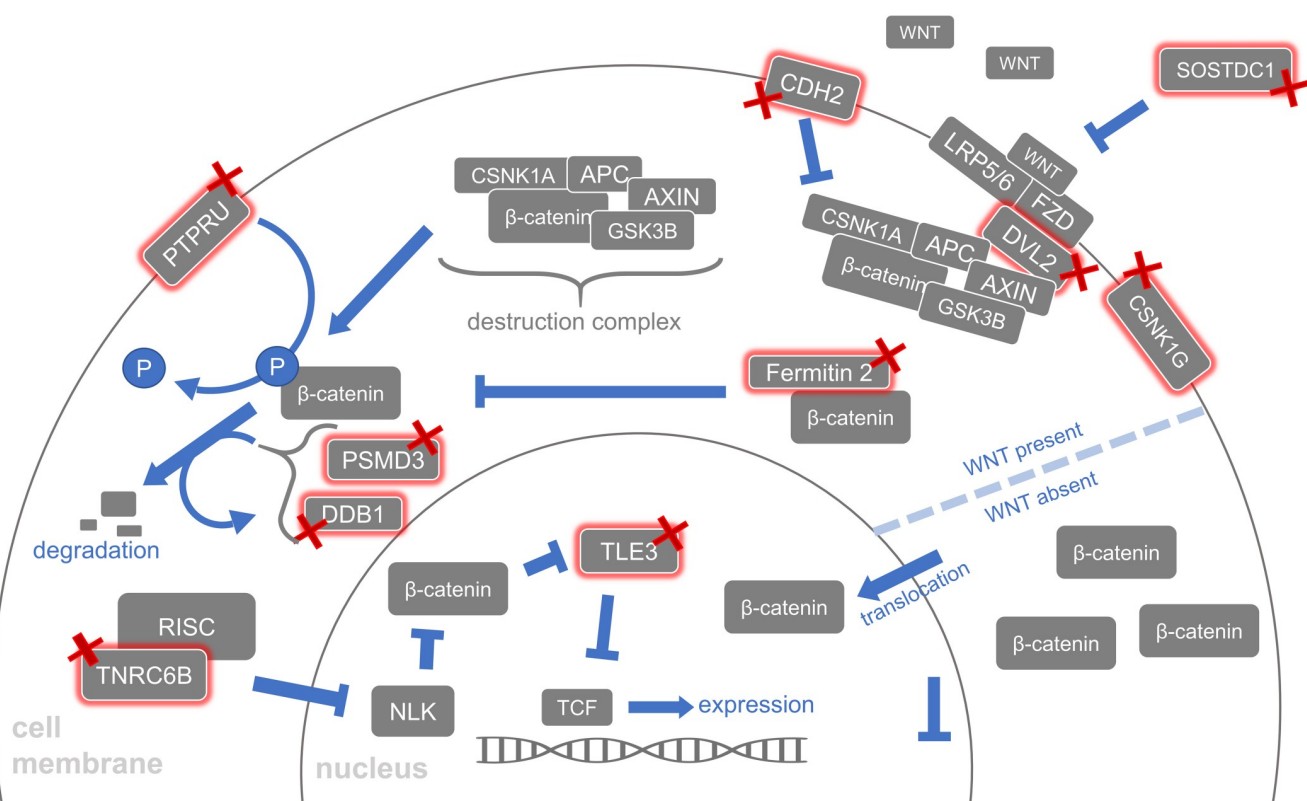

**Fig 5. Altered β-catenin.** A visual summary of the WNT/β-catenin cascade including genes disrupted in the myelomeningocele populations. The level of β-catenin, which is coded for by the human homolog *CTNNB1*, is regulated by many proteins in the WNT ligand's target cell. The proteins CSNK1G, DVL2, DDB1, PSMD3, PTPRU, Fermitin 2 (coded by *FERMT2*), CPE, PPP2R1A (not shown), and SOSTDC1 contribute to β-catenin's regulation and all have higher mutational burdens in one of the myelomeningocele populations compared to gnomAD references.

(FZD), then DVL is recruited and the PCP branch of WNT signaling is initiated (Fig 4). Our results reproduce and expand the findings of several current human NTD studies that show damaging variants in PCP genes play a role in the development of NTDs [8].

*DVL2*, a human homolog of the Dishevelled gene family, associated with myelomeningocele in the EA population. Dvl2 is a Dvl family protein essential in the PCP pathway [6]. Specifically, Dvl2 is required for endocytosis of the activated Wnt receptor, Frizzled [41]. Frog knockouts for the *DVL2* homolog *Xdsh* lack convergent extension and consequently display open neural tubes [42]. Similarly, mouse knockouts for *Dvl2* display a spina bifida phenotype [28]. Also, *DVL2* is differentially expressed in the caudal neural tube during neural tube closure [23]. Therefore, loss of correct *DVL2* function in some EA population subjects may cause myelomeningocele by preventing convergent extension during neural tube closure through disruption of PCP signaling.

*PRICKLE2* on chromosome 3 associated with myelomeningocele in the MA population. *PRICKLE2* is one of the two vertebrate homologs of the fruit fly's *Prickle* [43]. As summarized in Y. Yang & Mlodzik, 2015, *Prickle* and *Vangl* help establish PCP by directly antagonizing the formation of the Frizzled/FZD-Disheveled/DVL complex [44]. Because this polarity is required for convergent extension, loss of *PRICKLE2* may prevent proper convergent extension and thus failure of the neural tube to close. Another human myelomeningocele association study revealed more potential single-nucleotide polymorphism associations in *PRICKLE2* than any other gene among three of the four ethnicities evaluated; however, similar to our own

findings, the study failed to establish any strong associations to myelomeningocele in PCP genes [45]. Similarly, a targeted sequencing study of ninety human cranial NTD cases reported a discovery of likely damaging rare variants in *PRICKLE2* as well [46].

*CDH2*, which codes for a cadherin cell adhesion protein, associated with myelomeningocele in the MA population. Other cadherin genes such as *CELSR1* are expressed diversely in the developing neural tube [47]. Also, mutations in cell adhesion protein genes such as *Celsr1*, *EphrinA5*, and *EphA7* cause NTD phenotypes in mice [48, 49]. Moreover, *CDH2* itself is already implicated in the cause of NTDs because homozygous mouse knockouts for the homolog *Cdh2* displayed a wavy neural tube phenotype [25].

The gene *FUZ* on chromosome 19, whose mouse homolog *Fuz* is a PCP effector protein required for ciliogenesis by transporting DVL to the cilium [26, 27, 50], is also associated with myelomeningocele in the MA population. Five of the eight MA subjects with qualifying variants in *FUZ* possessed the variant NC_000019.9:g.50315872C>T p.(Ser78Asn), a variant which was 21 times more frequent in the MA cases than the matched gnomAD references. Murdoch and Copp reviewed the complex relationship between cilia and NTDs [51]. Similar to the cilia proteins associated with exencephaly, perhaps FUZ's transport of DVL to the cilium can also influence neural tube closure in myelomeningocele subjects. Indeed, mice with homozygous loss of *Fuz* expression display NTD phenotypes such as exencephaly before dying *in utero* [26, 27]. Also, multiple human myelomeningocele subjects possessed nonsynonymous mutations in *FUZ* that were not found in reference subjects, and these human variants revealed impaired cilia formation when tested in mouse cell lines [52].

## Myelomeningocele exomes with canonical WNT/β-catenin pathway genes disrupted

Out of the seventeen genes that nominally associated with risk for myelomeningocele, eleven were from the canonical WNT pathway. A high proportion in the canonical pathway is surprising, given that most WNT signaling genes previously associated with myelomeningocele are from the noncanonical PCP pathway, whose role in neural tube closure is more established. When a WNT family protein binds FZD and low-density lipoprotein receptor-related protein 5 or 6 (LRP5/6), the canonical WNT signaling pathway is activated (Fig 5). The pathway involves multiple events that influence the level of β-catenin in the target cell. If present in sufficient amount, β-catenin translocates to the nucleus where it displaces transducin-like enhancer/TLE family proteins from binding T-cell factor/TCF proteins and proceeds to upregulate the expression of downstream gene targets [53].

As mentioned in the section on PCP, *DVL2* associates with myelomeningocele in the EA population. In addition to their role in PCP, Dishevelled (Dvl/Dsh) proteins are important for canonical Wnt signaling, acting downstream of Wnt, Fzd, and Lrp5/6 [54, 55]. *Dvl2* is already implicated in the cause of myelomeningocele, because two to three percent of *Dvl2* double knockout mice display spina bifida phenotype [28]. In humans, *DVL2* is also differentially expressed in the caudal human neural tube during closure [23].

*DDB1*, which codes for UV damage-specific DNA-binding protein 1, associated with subjects in the EA population as well. DDB1 is used as an adaptor protein for the ubiquitin ligase protein, CUL4 [56]. Other E3 ubiquitin ligases from this family (CUL1 and CUL3) target β-catenin and Disheveled for degradation [57]. Although more research is needed, it is possible that a nonfunctional *DDB1* gene might decrease β-catenin degradation, consequently increasing the β-catenin signal in WNT-targeted cells. *DDB1* is particularly interesting, because the Fisher's exact *P* value for its mutational burden analysis almost passed the strict Bonferroni correction threshold adjusted for the 189 genes compared in the populations of European

ancestry (7.88E-4 versus 2.65E-4) and *DDB1* is differentially expressed in the caudal human neural tube during closure [23]. In mice, *Ddb1* null embryos were extensively degenerated and died at early embryonic stage. Conditional inactivation of *Ddb1* in the brain causes neural progenitor cells to apoptose, leading to neuronal degeneration, brain hemorrhages, and neonatal death [58]. However, *DDB1* has not been previously associated with myelomeningocele.

As mentioned in the section on WNT trafficking, *CPE* associated with myelomeningocele in the MA population with one qualifying variant found in half of the relevant subjects. Whereas full length CPE interacts with the Wnt ligand, *CPE*'s splice variant CPE-ΔN localizes to the nucleus, increases β-catenin expression, and induces expression of Wnt target genes [40]. Plausibly, loss of CPE-ΔN might indirectly lower gene expression driven by β-catenin. Though *CPE* has not been previously associated with myelomeningocele, *CPE* is differentially expressed in the caudal neural tube during human neural tube closure [23].

*PPP2R1A* on chromosome 19 also associated with myelomeningocele in the MA population. Two of the three variants that created the association were at the same location in two of three MA subjects carrying the qualifying variants. The variant NC_000019.9:g.52725413G>T p.(Arg527Leu) was over 34 times more frequent in the MA population than in the gnomAD AMR reference population and the variant NC_000019.9:g.52725413G>A (p.(Arg527His) was not reported in the gnomAD AMR references. *PPP2R1A* codes for a conserved subunit of the heterotrimeric protein PP2A, a serine/threonine protein phosphatase [59, 60]. Another subunit, B56, may direct PP2A to down-regulate the expression of Wnt/β-catenin effector genes by decreasing the amount of β-catenin in the cell [61]. It is possible this occurs by directly complexing with Axin [62] and dephosphorylating part of the APC complex [61]. Given these understandings, loss of PPP2R1A function could lead to increased expression of β-catenin effector genes. *PPP2R1A* is another novel gene association with myelomeningocele.

*TNRC6B* codes for an Argonaut-associated RNA and shows an association with myelomeningocele in the MA population. The Argonaut molecule is a recognition motif-containing protein that participate in RNA interference [63]. The *TNRC6B* RNA serves as one component of a RISC complex that inhibits NLK translation [64] and the NLK protein participates in the WNT/β-catenin pathway by causing the dissociation of the β-catenin complex from DNA [65]. Importantly, *TNRC6B* is differentially expressed in closing caudal human neural tube [23]. So, it is possible that a nonfunctional *TNRC6B* transcript compromises the RISC complex and indirectly decreases β-catenin role in downstream gene expression in myelomeningocele subjects. Our study is the first to association *TNRC6B* and myelomeningocele.

*SOSTDC1* on chromosome 7, which codes for sclerostin domain-containing protein 1, associated with myelomeningocele in the EA population. The variant NC_000007.13: g.16505280G>A p.(Ala5Val) was found in two of three EA subjects who possessed qualifying variants and both subjects were heterozygous for this variant. The NC_000007.13: g.16505280G>A p.(Ala5Val) variant was not present in any of the gnomAD NFE reference subjects. In mouse tooth development, the homolog Sostdc1 serves as an inhibitor of Lrp5/6-mediated Wnt signaling [66], but *SOSTDC1* has not previously been associated with myelomeningocele.

## Gene ontology enrichment analysis

None of the nominally significant genes among MA myelomeningocele subjects overlapped with the genes from the EA subjects (Table 1). However, gene ontologies from each gene set did overlap between ethnicities. GO enrichment analysis revealed that the three GO terms titled "cell-cell signaling by wnt", "Wnt signaling pathway", and "Signaling by Wnt" were enriched in disrupted genes from both populations. Enrichment of these broad terms were

expected because the original 523 genes were retrieved using the overarching "cell-cell signaling by wnt" term. In other words, any subset of the original gene list is likely to be enriched for terms that describe the general WNT signaling pathway.

More interesting are the lower level, more specific GO terms that were enriched in both populations' signaling pathways. These shared ontologies suggest a shared mechanism behind myelomeningocele in these two ethnicities, despite having high mutational burdens in non-overlapping genes. The clearest example is negative regulation of the canonical WNT/β-catenin component of WNT signaling.

That said, many GO terms were only enriched in one or the other population's disrupted gene lists. So, while some mechanisms may be shared, others may be unique to each ethnicity. This is one possible explanation behind the two populations' non-overlapping sets of disrupted genes. The ontology terms enriched in only the MA gene list largely pertain to PCP, whereas the story is less clear for the EA ontology terms.

Another difference between populations can be seen in Fig 2, where MA subjects tended to have more disrupted genes than EA subjects. More research is needed, but the discrepancy could help explain why the prevalence of myelomeningocele among Mexicans is higher than in non-Hispanic whites [67].

## Limitations

The current study introduced 17 genes associated with risk for myelomeningocele with nominal significance (Fisher exact *P* values ≤ 0.05). Bonferroni correction for multiple comparisons was applied to each analysis using the number of compared genes as a denominator (173 in Hispanic comparisons and 189 in European ancestry comparisons) and 0.05 as the numerator alpha to calculate the correction threshold. While some genes came close, none of the *P* values fell below the correction threshold. A large enough group of myelomeningocele subjects would lend enough statistical power to achieve the strict statistical significance of a Bonferroni correction, but exome data for 511 myelomeningocele subjects is considerable, given the resource-intensive nature of gathering samples. The genes discussed above are suggested candidate genes.

Two of our quality control filters assume that each variant exists in Hardy-Weinberg equilibrium within the myelomeningocele population (inbreeding coefficient < -0.3, alternate allele depth ≥ 25%). However, if a selective pressure acts on a variant locus, that Hardy-Weinberg assumption is not met. Therefore, our analysis may exclude important variant loci that are under intense selective pressure.

A new approach was taken by limiting our analysis to a subset of human genes rather than evaluating the entire exome at once. The high concentration of current NTD candidate genes within the WNT signaling pathway and the relevance of WNT signaling to neural tube closure mechanisms such as convergent extension prompted this focused approach. A more inclusive approach would also be valuable.

## Perspective

Authors of a recent review list genes containing deleterious rare variants that have been associated with NTDs in humans. This study proffers *ANKRD6*, *CELSR1*, *CELSR2*, *CELSR3*, *DVL2*, *DVL3*, *FZD*, *PK1*, *VANGL1*, *VANGL2*, *LRP6*, *PTK7*, and *SCRIB1* as associated genes [68]. Except for *DVL2*, our study may appear not to corroborate findings in their review. We offer three explanations for this discrepancy. First, the current study uniquely employs a gene-based mutational burden analysis. Adopting a similar methodology to those studies summarized in the review may yield similar results. Second, our subjects differ in ethnicity from many of the

populations discussed in other studies. As the current study suggests, ancestry may influence which genes associate with myelomeningocele. Third, any variants that were not equally covered in both case and reference populations were filtered out before analysis. If, for example, an important variant was discovered in the myelomeningocele subjects while the variant's location was not covered in the corresponding gnomAD reference population, that variant would not contribute to the mutational burden analysis.

In summary, we report seventeen genes within the known WNT signaling pathways which may play a role in the development of myelomeningocele. As discussed above, the genes *PORCN*, *DVL2*, *CDH2*, *FUZ* are already suspected to play a role in NTD development from studies in animal models and in some cases humans, however the remaining thirteen genes reported here are new in their possible association with myelomeningocele.

## Supporting information

**S1 Fig. Calcium WNT pathway.** A depiction of PLCB1's role in the noncanonical $Ca^{2+}$ WNT signaling pathway. PLCB1 activates the G-protein coupled receptor's alpha subunit, which is necessary for the downstream activation of CAMK2A.
(TIF)

**S1 Table. Lists of GO-retrieved WNT signaling genes.**
(XLS)

**S2 Table. Hispanic gene-based mutational burden analysis results.**
(XLS)

**S3 Table. European Ancestry gene-based mutational burden analysis results.**
(XLS)

**S4 Table. Hispanic mutational burden analysis variants.** Variant data for those variants used in the Hispanic mutational burden analysis which fall within nominally significant disrupted genes.
(XLS)

**S5 Table. European ancestry mutational burden analysis variants.** Variant data for those variants used in the European ancestry mutational burden analysis which fall within nominally significant disrupted genes.
(XLS)

**S6 Table. RDV gene combinations.** A list of nominally significant disrupted gene combinations that contained RDVs in the same sample.
(XLSX)

## Acknowledgments

We give special thanks to Lawrence C. Shimmin and Do-Kyun Kim from the Department of Epidemiology, Human Genetics and Environmental Sciences, School of Public Health at The University of Texas Health Science Center at Houston (UTHealth) for their work and expertise in WES. Special thanks to Karen Soldano and Melanie Garett in Department of Medicine at Duke University for their technical support and providing human neural tubes RNA SAGE information to annotate WES results.

## Author Contributions

**Conceptualization:** Luke Hebert, Paul Hillman, Craig Baker, Hope Northrup, Kit Sing Au.

**Data curation:** Michael Brown, Allison Ashley-Koch, James E. Hixson, Alanna C. Morrison, Hope Northrup, Kit Sing Au.

**Formal analysis:** Luke Hebert, Paul Hillman, Craig Baker, Michael Brown, Kit Sing Au.

**Funding acquisition:** Hope Northrup, Kit Sing Au.

**Investigation:** Luke Hebert, Paul Hillman, Craig Baker, Kit Sing Au.

**Methodology:** Luke Hebert, Paul Hillman, Craig Baker, Michael Brown, James E. Hixson, Kit Sing Au.

**Project administration:** Hope Northrup, Kit Sing Au.

**Resources:** Hope Northrup, Kit Sing Au.

**Software:** Luke Hebert, Paul Hillman, Craig Baker, Michael Brown.

**Supervision:** Alanna C. Morrison, Hope Northrup, Kit Sing Au.

**Validation:** James E. Hixson, Kit Sing Au.

**Visualization:** Luke Hebert, Paul Hillman, Craig Baker.

**Writing – original draft:** Luke Hebert.

**Writing – review & editing:** Luke Hebert, Paul Hillman, Craig Baker, Michael Brown, Allison Ashley-Koch, James E. Hixson, Alanna C. Morrison, Hope Northrup, Kit Sing Au.

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
