## [Decision Letter · Decision Letter 0]

15 Jul 2020

PONE-D-20-18473

Burden of Rare Deleterious Variants in WNT Signaling Genes Among 511 Myelomeningocele Patients

PLOS ONE

Dear Dr. Au,

Thank you for submitting your manuscript to PLOS ONE. After careful consideration, we feel that it has merit but does not fully meet PLOS ONE’s publication criteria as it currently stands. Therefore, we invite you to submit a revised version of the manuscript that addresses the points raised during the review process.

We look forward to receiving your revised manuscript.

Kind regards,

Zoha Kibar

Academic Editor

PLOS ONE

Journal Requirements:

2. Please provide additional details regarding participant consent.

In the ethics statement in the Methods and online submission information, please ensure that you have specified (i) whether consent was informed and (ii) what type you obtained (for instance, written or verbal, and if verbal, how it was documented and witnessed).

If the need for consent was waived by the ethics committee, please include this information.

Reviewers' comments:

Reviewer's Responses to Questions

**Comments to the Author**

1. Is the manuscript technically sound, and do the data support the conclusions?

Reviewer #1: Partly

Reviewer #2: Yes

2. Has the statistical analysis been performed appropriately and rigorously? 

Reviewer #1: Yes

Reviewer #2: I Don't Know

3. Have the authors made all data underlying the findings in their manuscript fully available?

Reviewer #1: Yes

Reviewer #2: Yes

4. Is the manuscript presented in an intelligible fashion and written in standard English?

Reviewer #1: Yes

Reviewer #2: Yes

5. Review Comments to the Author

Reviewer #1: Summary: The authors of this manuscript have identified that specific genes in the WNT signaling pathway have a higher mutational burden in terms of the pathogenesis of myelomeningocele, a neural tube defect in humans. The authors developed an extensive analysis pipeline to uncover candidate genes. Ten genes were discovered that were primarily important for planar cell polarity with higher mutational burden in the population of Mexican Americans, and seven genes with higher mutational burden in the population of European ancestry. Then the authors performed GO analysis and established the relationship between the rare deleterious variants (RDVs) in genes within different aspects of the Wnt signaling pathway. This study may be helpful to understand the genetic pathogenesis of human neural tube defects.

There are two major caveats: one, the same rare mutations are also found in control populations and two, the absence of experimental evidence that these mutations damage biological functions. This weakens the conclusions for pathogenesis of these variations.

Major concerns:

1. In the introduction, the authors could better outline their thought process of how they went from stating that there are 372 disrupted genes in mouse models of NTDs, to saying they suspect these deleterious mutations for human NTDs could be in the WNT signaling pathway above any other cell signaling pathway important for development.

2. In the current cases, are there clinical phenotypes other than myelomeningocele?

3. If possible, please describe the biological context of the gnomAD controls. Were they overall “healthy” with no other health conditions?

4. Page 10 line 211, what do the authors mean by two European populations? Please describe the two populations clearly.

5. How do the authors explain that nominally significant genes associated with myelomeningocele do not overlap in the EA and MA populations?

6. Provide a table of the 17 genes indicating which ones are heterozygous and which ones are homozygous in the patients. How many frameshift mutations were found in the present cohort? Were they included in the statistical analysis?

7. Fig. 2 indicates there may be more than one Wnt pathway gene that contains an RDV in an individual case. Please include a table that shows which genes (2 or more) that contain RDVs in an individual. Also include whether the NTD is more severe or associated with other phenotypes in these cases with multiple Wnt RDVs. Are multiple Wnt RDVs observed in controls? Do the multiple RDV mutations affect proteins within a complex or do they hit different aspects of the Wnt pathway?

8. It looks like all RDVs reported in the present cohorts could be found in control population, and there are no functional experiments to demonstrate the RDVs can disrupt the biological functions, so it is problematic to evaluate the contribution of current RDVs to the pathogenicity of myelomeningocele.

9. Regarding the GO analysis, since the basis of the study is to examine genes functioning in the WNT signaling pathway, including core genes and associated genes, it is unreasonable to do GO analysis and draw conclusion on the WNT signaling pathway, although the deeper analysis within the pathway relative to population RDV is OK. However, some genes like DVL2, which is a hub of Wnt signaling and is involved in both canonical and non-canonical Wnt signaling, are categorized in Table 2 as only within the canonical signaling pathway.

10. How were miRNA binding sites in the genome defined?

11. Line 240, please explain SCF-KIT signaling.

12. How do the authors consider the presence of RDV mutations in genes that are not detected in the neural tissue of humans (Carnegie stage 12 and stage 13, as cited)?

13. The OR value may not be appropriate to evaluate risk of disease, as the number of control and case are relatively mismatched.

14. Some papers have already reported the genetic contributions of genes functioning in Wnt signaling pathways to neural tube defects, please summarize them in the discussion and clarify the ethnicity feature of genetic mutations of Wnt signaling pathway in human neural tube defects.

Minor concerns:

15. Lines 240-243 please expand on these signaling pathways mentioned and how they are related to the WNT signaling pathway genes being affected. If they aren’t related, state that or take it out of this section and put it in the discussion where you can discuss their possible importance.

16. is there a DDB1 mouse mutant? What is its phenotype?

17. Page 5 line103 and 107, “S1” should be corrected to Table S1?

18. Page 7 line 150, a CADD paper should be cited here.

19. In Table 1 “CI” means 95% CI or something else?

Reviewer #2: The authors hypothesized that some genes in the WNT signaling network have a higher mutational burden in myelomeningocele (MMC) subjects than in controls. Exome sequencing was performed from 511 MMC was performed and data from 29,940 ethnically matched subjects was provided by version 2 of the publicly available Genome Aggregation Database. To compare mutational burden, the authors searched rare deleterious variants across each of 523 human WNT signaling genes in case and control populations.

10 WNT signaling genes were disrupted with a higher mutational burden among Mexican

American MMC subjects compared to controls and 7 genes were disrupted among individuals of European ancestry compared to controls. Gene ontology enrichment analyses indicate that genes disrupted only in the Mexican American population play a role in PCP pathway whereas genes identified in both populations are important for the regulation of canonical WNT signaling

The authors presented an impressive amount of data on a very large group of patients and mutational burden analysis provides more power than single-variant approach.

6. PLOS authors have the option to publish the peer review history of their article (what does this mean?). If published, this will include your full peer review and any attached files.

Reviewer #1: No

Reviewer #2: No

---

## [Author Response · Author response to Decision Letter 0]

15 Aug 2020

We appreciate the editor's email clarifying that authors do not need to response to Reviewer#1's comments on present of some rare variants in control populations and lack of experimental evidence that the rare variants damage biological functions.

---

## [Decision Letter · Decision Letter 1]

31 Aug 2020

Burden of Rare Deleterious Variants in WNT Signaling Genes Among 511 Myelomeningocele Patients

PONE-D-20-18473R1

Dear Dr. Au,

We’re pleased to inform you that your manuscript has been judged scientifically suitable for publication and will be formally accepted for publication once it meets all outstanding technical requirements.

Kind regards,

Zoha Kibar

Academic Editor

PLOS ONE

Additional Editor Comments (optional):

Reviewers' comments:

Reviewer's Responses to Questions

**Comments to the Author**

1. If the authors have adequately addressed your comments raised in a previous round of review and you feel that this manuscript is now acceptable for publication, you may indicate that here to bypass the “Comments to the Author” section, enter your conflict of interest statement in the “Confidential to Editor” section, and submit your "Accept" recommendation.

Reviewer #1: All comments have been addressed

2. Is the manuscript technically sound, and do the data support the conclusions?

Reviewer #1: (No Response)

3. Has the statistical analysis been performed appropriately and rigorously? 

Reviewer #1: (No Response)

4. Have the authors made all data underlying the findings in their manuscript fully available?

Reviewer #1: (No Response)

5. Is the manuscript presented in an intelligible fashion and written in standard English?

Reviewer #1: (No Response)

6. Review Comments to the Author

Reviewer #1: (No Response)

7. PLOS authors have the option to publish the peer review history of their article (what does this mean?). If published, this will include your full peer review and any attached files.

Reviewer #1: No

---

## [Editor Report · Acceptance letter]

16 Sep 2020

PONE-D-20-18473R1 

Burden of Rare Deleterious Variants in WNT Signaling Genes Among 511 Myelomeningocele Patients 

Dear Dr. Au:

I'm pleased to inform you that your manuscript has been deemed suitable for publication in PLOS ONE. Congratulations! Your manuscript is now with our production department. 

Kind regards, 

on behalf of

Dr Zoha Kibar 

Academic Editor

PLOS ONE